# The Transcription Factor OVOL2 Represses ID2 and Drives Differentiation of Trophoblast Stem Cells and Placental Development in Mice

**DOI:** 10.3390/cells9040840

**Published:** 2020-03-31

**Authors:** Mariyan J. Jeyarajah, Gargi Jaju Bhattad, Dendra M. Hillier, Stephen J. Renaud

**Affiliations:** 1Department of Anatomy and Cell Biology, Schulich School of Medicine and Dentistry, University of Western Ontario, London, ON N6A5C1, Canada; mjeyaraj@uwo.ca (M.J.J.);; 2Children’s Health Research Institute, London, ON N6C2V5, Canada; 3Lawson Health Research Institute, London, ON N6C2R5, Canada

**Keywords:** embryogenesis, blastocyst, transcriptional repression, trophoblast stem cells, trophoblast differentiation, placenta, OVO-like 2

## Abstract

Trophoblasts are the first cell type to be specified during embryogenesis, and they are essential for placental morphogenesis and function. Trophoblast stem (TS) cells are the progenitor cells for all trophoblast lineages; control of TS cell differentiation into distinct trophoblast subtypes is not well understood. Mice lacking the transcription factor OVO-like 2 (OVOL2) fail to produce a functioning placenta, and die around embryonic day 10.5, suggesting that OVOL2 may be critical for trophoblast development. Therefore, our objective was to determine the role of OVOL2 in mouse TS cell fate. We found that OVOL2 was highly expressed in mouse placenta and differentiating TS cells. Placentas and TS cells lacking OVOL2 showed poor trophoblast differentiation potential, including increased expression of stem-state associated genes (*Eomes, Esrrb, Id2*) and decreased levels of differentiation-associated transcripts (*Gcm1*, *Tpbpa, Prl3b1, Syna*). Ectopic OVOL2 expression in TS cells elicited precocious differentiation. OVOL2 bound proximate to the gene encoding inhibitor of differentiation 2 (ID2), a dominant negative helix-loop-helix protein, and directly repressed its activity. Overexpression of ID2 was sufficient to reinforce the TS cell stem state. Our findings reveal a critical role of OVOL2 as a regulator of TS cell differentiation and placental development, in-part by coordinating repression of ID2.

## 1. Introduction

Trophoblast is the first cell type to become specified during early embryogenesis. These cells ultimately form the epithelial portion of the placenta and perform crucial functions that support fetal growth and development. The precursors for all distinct trophoblast lineages are trophoblast stem (TS) cells, which can be derived from the early embryo and are a powerful tool for determining cell fate decisions in extraembryonic lineages. Precise spatial and temporal control of TS cell differentiation is fundamental for normal placental morphogenesis and function [1], but the regulation of TS cell differentiation is poorly understood.

Mouse models have been instrumental in advancing our understanding of TS cell development and providing insight into the morphology and function of the placenta [2,3]. The regulation of mouse TS cell maintenance and differentiation has been extensively studied, since culture conditions that facilitate long-term TS cell culture and differentiation in this species have been known for several decades, whereas culture conditions for human TS cells have only recently been described [4,5]. Additionally, our understanding of the complexities of TS cell development and placental organization has greatly benefitted from mouse mutagenesis, where placentation defects are implicated in a considerable proportion of mouse mutants exhibiting embryonic lethality [6]. Many genes essential for rodent trophoblast development have subsequently been shown to have critical roles in human placentation, or are dysregulated during pregnancy pathologies, demonstrating the value of rodent models for enhancing our understanding of placental development.

During mouse TS cell differentiation, a subset of genes associated with the stem-state are repressed, including caudal box homeodomain 2 (*Cdx2*), estrogen related receptor beta (*Esrrb*), eomesodermin (*Eomes*), and inhibitor of DNA binding 2 (also called inhibitor of differentiation 2, *Id2*). Additionally, expression of genes associated with differentiation progressively increase, including placental lactogen 2 (*Prl3b1*), glial cells missing homolog 1 (*Gcm1*), syncytin a (*Syna*), syncytin b (*Synb*), achaete-scute family bHLH transcription factor 2 (*Ascl2*), and trophoblast-specific protein alpha (*Tpbpa*). These molecular changes are precisely controlled in time and space, enabling differentiated trophoblasts to contribute to specialized placental regions called the junctional zone (comprised of trophoblast giant cells, spongiotrophoblasts, and glycogen cells), and labyrinth zone (which contains sinusoidal giant cells and two syncytiotrophoblast layers) [7]. Each cell type is responsible for tasks that are critical for placental function. For instance, trophoblast giant cells facilitate implantation and attachment of the blastocyst to the uterine epithelium, as well as secretion of various hormones, which are responsible for regulating development of both maternal and fetal compartments [8]. Glycogen cells differentiate into an invasive phenotype, which migrate into the decidua and assist in remodeling the maternal blood vessels so that the placenta is provided with an adequate supply of maternal blood throughout gestation [8]. The syncytiotrophoblast layers contact maternal blood and form an exchange surface to facilitate delivery of nutrients and oxygen into the fetal circulation. Disrupting the formation of these cell types results in a malformed placenta and can be detrimental to the health of the fetus [7,9]. Thus, a better understanding of the mechanisms controlling TS cell development and differentiation is necessary.

A family of OVO-like (OVOL) transcription factors is highly implicated in epithelial development [10]. In mice and humans, there are three OVOL paralogs—*Ovol1*/*OVOL1*, *Ovol2*/*OVOL2*, and *Ovol3*/*OVOL3*—that exhibit homology to *Drosophila ovo/shavenbaby*. The *OVOL2* gene additionally encodes three major isoforms (OVOL2A, OVOL2B, and OVOL2C) through distinct transcription start sites and alternative splicing, which differ in transcriptional activity [11]. OVOL factors contain four conserved C2H2 zinc finger domains at their C-terminal ends that facilitate DNA binding, but differ in the structure of their N-terminal regions [12]. OVOL factors regulate differentiation of myriad epithelial cell types including epidermis [13], hair follicles [14], kidney, spermatocytes [15], and mammary epithelial cells [16]. Mutagenesis of *Ovol1* and *Ovol2* in mice has revealed some functional redundancy as well as some critical tissue-specific differences. Notably, mice lacking OVOL1 die postnatally due to dysregulated epidermis and kidney development [17]. Conversely, OVOL2 deficiency causes embryonic lethality around embryonic day (E)10.5 resulting from major defects in blood vessel expansion, heart formation, and placental development [18].

We previously demonstrated a critical role for OVOL1 in human syncytiotrophoblast formation. However, we did not detect OVOL1 in mouse or rat placental tissue or in rodent TS cells [19]. Rodent placentas and TS cells, on the other hand, possessed high levels of OVOL2. The high expression of OVOL2 in rodent placentas, fundamental role of OVOL factors for epithelial cell fate decisions, and mid-gestation embryonic lethality with placentation defects observed in *Ovol2*-mutant mice prompted us to investigate the functional role of OVOL2 for mouse TS cell development and placental formation. Herein, we found a critical role for OVOL2 in TS cell differentiation and placental development in mice. We also uncovered a novel pathway through which OVOL2 functions to facilitate TS cell differentiation, involving repression of the gene encoding ID2.

## 2. Materials and Methods

### 2.1. Animals

Male and female C57BL/6N and CD1 mice were obtained from Charles River Laboratories (Wilmington, MA, USA). *Ovol2^+/-^* mice were previously described [18], and were generously provided by Seiji Ito through RIKEN Bioresource Research Centre (Tsukuba, Ibaraki Prefecture, Japan) based on a Materials Transfer Agreement. All mice were maintained in a 12 h light: 12 h dark cycle with food and water available *ad libitum*. To induce pregnancy, males and females were placed into the same cage overnight, and the presence of a seminal plug the following morning was designated E0.5. Protocols involving the use of mice were approved by the University of Western Ontario Animal Care and Use Committee, which was approved in March 2017 (protocol 2017-012).

### 2.2. Experimental Protocol and Tissue Collection

To evaluate expression patterns of OVOL2 in various fetal and placental tissues throughout gestation, pregnant C57BL/6N mice were sacrificed on E9.5, E12.5, E15.5, and E18.5. Placentas and embryos were collected at each time point, then immersed in dry ice-cooled heptane for immunohistochemical analysis, or snap-frozen in liquid nitrogen for molecular analyses, and stored at −80 °C.

To determine the importance of OVOL2 for placental development, placental morphology was evaluated in *Ovol2^+/+^, Ovol2^+/-^*, and *Ovol2^-/-^* conceptuses. *Ovol2^+/-^* males were bred with wild-type C57BL/6N females to maintain a stock of heterozygotes. To genotype pups, DNA was extracted from tail clips using REDExtract-N-Amp Tissue PCR Kit (Sigma-Aldrich, Oakville, ON, Canada), according to the manufacturer’s protocol. Polymerase chain reaction (PCR) was performed using DreamTaq DNA polymerase (ThermoFisher Scientific, Mississauga, ON, Canada) and primers spanning the *Ovol2* deletion site (Table 1). PCR conditions were as follows: initial holding step (95 °C for 3 min), followed by 32 cycles of PCR (95 °C for 30 s, 63 °C for 30 s, and 72 °C for 30 s), and a final elongation phase at 72 °C for 12 min. To obtain *Ovol2^+/+^, Ovol2^+/-^,* and *Ovol2^-/-^* conceptuses, *Ovol2^+/-^* females were bred with *Ovol2^+/-^* males. Embryos and placentas were collected on E9.5. Embryos were genotyped, and placentas were either preserved in dry ice-cooled heptane or snap-frozen in liquid nitrogen, and stored at −80 °C.

For experiments involving generation of blastocyst-derived TS cells, *Ovol2^+/-^* females were bred with *Ovol2^+/-^* males. On E3.5, blastocysts were collected by flushing uterine horns with M2 media (Sigma Aldrich). TS cells were then derived from blastocysts, as described below. Mouse embryonic fibroblasts (MEFs), which are required for TS cell culture, were prepared from E12.5 fetal tissue as previously described [20]. Fetal tissue for preparation of MEFs was collected from CD-1 mice.

### 2.3. Cells

Media components for cell culture were obtained from Sigma-Aldrich unless otherwise specified, and all cells were maintained in a temperature and gas-controlled incubator at 37 °C, 5% CO_2_. MEFs were maintained in Dulbecco’s Modified Eagle Medium (DMEM) supplemented with 10% fetal bovine serum (FBS, ThermoFisher Scientific), 100 units/mL penicillin, and 100 μM streptomycin. To inhibit MEF proliferation, cells were treated with 5 μg/mL mitomycin C (M0503, Sigma-Aldrich) for 2 h.

Mouse TS cells (F4 line) were generously provided by Janet Rossant, and were also generated in-house following collection of blastocysts from *Ovol2*^+/-^ × *Ovol2*^+/-^ matings. TS cells were generated by placing blastocysts onto cell culture plates containing mitotically inactivated MEFs in the presence of TS cell media (Roswell Park Memorial Institute (RPMI)-1640 media supplemented with 20% FBS, 25 ng/mL fibroblast growth factor 4 (235-F4-025, FGF4, R&D systems, Minneapolis, MN, USA), 1 µg/mL heparin, 10 ng/mL activin A (338-AC-10, R&D systems), 100 μM 2-mercaptoethanol, 100 mM sodium pyruvate, 100 units/mL penicillin, and 100 μM streptomycin). Once large aggregates of TS cells appeared in culture, cells were separated by mechanical dissociation, and passaged. MEFs were gradually depleted from co-cultures by plating TS cells on gelatin-coated tissue culture plates. Subsequently, cells were maintained in TS cell media, 70% of which was preconditioned by MEFs. DNA was isolated from TS cells (Qiagen, Toronto, ON, Canada), and their genotype determined as described above.

TS cells were induced to differentiate by removal of MEF preconditioned media, FGF4, heparin, and activin A (differentiation media) [20]. To stimulate TS cell differentiation toward the syncytiotrophoblast II lineage (a sublineage of the labyrinth zone), cells were cultured in differentiation media, along with a GSK3β inhibitor, CHIR99021 (3 µM, Stemgent, Cambridge, MA, USA) [21]. Controls for TS cells exposed to CHIR99021 consisted of cells exposed to dimethyl sulfoxide (DMSO, 1:1000), which was the vehicle used to prepare CHIR99021.

HEK-293T cells were maintained in DMEM supplemented with 10% FBS, 100 units/mL penicillin, and 100 µM streptomycin. Cells were passaged by light trypsinization prior to reaching confluency.

### 2.4. Immunohistochemistry

Frozen mouse conceptuses were embedded in O.C.T. compound, cryosectioned at 10 μm, and fixed in 4% paraformaldehyde. Sections were then treated with 0.3% hydrogen peroxide in methanol to block endogenous peroxidases, permeabilized using 0.3% Triton-X, and blocked with 10% normal goat serum (ThermoFisher Scientific). Sections were immersed in a primary antibody specific for cytokeratin (628602, 1:400, BioLegend, San Diego, CA, USA), and then incubated with a mouse biotinylated secondary antibody, followed by Vectastain peroxidase (Vector Laboratories, Burlingame, CA, USA). Placental sections were then treated with an AEC Chromogen Solution (ThermoFisher Scientific), nuclei counterstained with hematoxylin, and sections mounted using Fluoromount-G (Southern Biotech, Birmingham, AL, USA). For fluorescent staining, sections were fixed and permeabilized as described above, and incubated with antibodies specific for TPBPA (104401, 1:100, Abcam, Toronto, ON, Canada), or cytokeratin conjugated to Alexa 488 (628608, 1:400, BioLegend). Sections immersed in antibodies specific for TPBPA were washed and then incubated with Alexa 555-conjugated anti-rabbit secondary antibodies. All sections were counterstained with 4′,6-diamidino-2-phenylindole (DAPI, ThermoFisher Scientific), mounted, and imaged using a Nikon DS-Qi2 microscope.

### 2.5. In Situ Hybridization

In situ hybridization was conducted using RNAScope, as per the manufacturer’s instructions (Advanced Cell Diagnostics, Newark, CA, USA). Briefly, sections were hydrated through a graded series of ethanol washes, subjected to peroxidase and protease treatment, and hybridized with probes specific to mouse *Ovol2*. Additional sections treated with probes specific for mouse *Ppib* and bacterial *dapB* served as positive and negative controls, respectively. Sections were then subjected to a series of amplification steps, treated with 3,3′-diaminobenzidine chromogen solution, and nuclei were counterstained with hematoxylin. Sections were dehydrated using increasing concentrations of ethanol, cleared using xylene, mounted with Cytoseal (ThermoFisher Scientific), and imaged using a Nikon DS-Qi2 microscope.

### 2.6. RT-PCR and Quantitative RT-PCR

RNA was extracted from cells and tissue using Ribozol (VWR International, Mississauga, ON, Canada), according to the manufacturer’s instructions, and converted into cDNA via reverse transcription (High Capacity cDNA kit, ThermoFisher Scientific). cDNA was diluted 1:10, then subjected to conventional PCR or quantitative PCR using primers detailed in Table 1. Conventional PCR was performed using DreamTaq DNA Polymerase (ThermoFisher Scientific). Cycling conditions involved an initial holding step (95 °C for 3 min), followed by 32 cycles of PCR (95 °C for 30 s, 58–63 °C for 30 s, and 72 °C for 30 s), and a final elongation phase at 72 °C for 12 min. Quantitative RT-PCR was performed using a CFX96 Touch (Bio-Rad Laboratories, Mississauga, ON, Canada) and Sensifast SYBR Green PCR Master Mix (FroggaBio, Toronto, ON, Canada). Cycling conditions involved an initial holding step (95 °C for 10 min), followed by 40 cycles of a two-step PCR (95 °C for 15 s and 60 °C for 1 min) and a dissociation phase. Relative mRNA expression was calculated using the ΔΔCt method, using the geometric mean from two constitutively expressed genes, *Rn18s* and *Ywhaz*, as reference RNAs.

### 2.7. Western Blotting

Cell and tissue lysates were prepared by immersing in radioimmunoprecipitation assay (RIPA) lysis buffer (50 mM Tris, 150 mM NaCl, 1% NP40, 0.5% sodium deoxycholate, 0.1% sodium dodecyl sulfate (SDS)) supplemented with protease inhibitor cocktail (Sigma-Aldrich). A modified bicinchoninic acid assay (Bio-Rad Laboratories) was used to measure protein concentrations. Approximately 40 μg of cell lysate was mixed with 4× reducing loading buffer (0.25 M Tris, 8% SDS, 30% glycerol, 0.02% bromophenol blue, 0.3 M dithiothreitol), boiled for 5 min, and subjected to SDS-polyacrylamide gel electrophoresis. Proteins were transferred to polyvinylidene difluoride membranes and probed using antibodies for OVOL2 (GTX120220, 1:500, GeneTex, Irvine, CA, USA), ACTB (47778, 1:1000, Santa Cruz Biotechnology, Santa Cruz, CA, USA), and FLAG (F1804, 1:1000, Sigma-Aldrich). Membranes were then incubated for 1 h with species-appropriate secondary antibodies, and signals detected using a LI-COR Odyssey imaging system (LI-COR Biosciences, Lincoln, NE, USA).

### 2.8. Transfection

To ectopically express FLAG-tagged OVOL2 and ID2 in mouse TS cells, pCMV6-Entry constructs containing the complete cDNA sequence of *Ovol2* and *Id2,* each linked to a 3′-sequence encoding a C-terminal FLAG tag, were obtained from OriGene (MR203678 and MR200792, respectively, Rockville, MD, USA). Plasmids (pCMV6-Empty and pCMV6-OVOL2) were transfected into HEK293T cells using Lipofectamine 2000 (ThermoFisher Scientific), as described previously [22], to verify specificity of the OVOL2 antibody used for Western blotting. *Ovol2* and *Id2* cDNAs were PCR amplified and cloned into a pEF-GFP vector (Addgene catalog 11154, a generous gift from Connie Cepko) downstream of an EF1α promoter. Mouse TS cells were transfected using jetPRIME transfection reagent (Polyplus Transfection, New York, NY, USA) with 2 µg of either pEF-GFP (CTRL), pEF-OVOL2-FLAG, or pEF-ID2-FLAG vectors, according to the manufacturer’s instructions. Briefly, plasmids were added to 200 µL jetPRIME buffer, mixed with 8 µL jetPRIME transfection reagent, and incubated for 10 min. Following incubation, the transfection mix was added dropwise to TS cells in TS cell media. Quantitative RT-PCR and/or Western blotting were performed as described above to determine successful transfection of *Ovol2* or *Id2*.

### 2.9. EdU Cell Proliferation Assay

A 5-ethynyl-2′-deoxyuridine (EdU) incorporation assay was conducted to assess TS cell proliferation and was performed according to the manufacturer’s instructions (Click-iT EdU Cell Proliferation Assay, ThermoFisher Scientific). Briefly, 10 µM EdU was added to TS cell media, and provided to cells for 72 h. Following incubation, mouse TS cells were fixed in 4% paraformaldehyde, permeabilized using 0.3% Triton-X, and incubated with Click-iT EdU reaction cocktail for 30 min. Nuclei were stained using Hoechst 33342. Cells were imaged using a Zeiss Axio fluorescence microscope.

### 2.10. Chromatin Immunoprecipitation

Chromatin Immunoprecipitation (ChIP) was performed as previously described [19]. Mouse TS cells (F4 line) were differentiated for three days, fixed for 10 min with 0.7% formaldehyde, lysed, and sonicated using a Bioruptor (Diagenode, Denville, NJ, USA) to generate DNA fragments less than 400 bp. 1% of sonicated nuclear lysate was removed to serve as an input control. Half of the remaining lysate was incubated with OVOL2 antibody (5 μg, GTX120220, GeneTex); the other half was incubated with a negative control rabbit IgG (5 μg, 2729, Cell Signaling Technology, Danvers, MA, USA) overnight at 4 °C. Immunoprecipitated chromatin fragments were then captured using protein G-conjugated Sepharose beads (Sigma-Aldrich). Following capture, DNA fragments were eluted and assessed by quantitative PCR. Values were normalized relative to input. Primers specific for regions of DNA that span the consensus OVOL2 DNA-binding motif, CCGTTA [12], are found in Table 2.

### 2.11. Luciferase Assay

A section of the mouse *Id2* promoter (from -1420 to +153) containing a putative OVOL2 binding site was cloned upstream of firefly luciferase into a pGL2-Luc vector (E1641, Promega, Madison, WI, USA). HEK-293T cells were transfected using Lipofectamine 2000 (ThermoFisher), as previously described [22], with 100 ng of pGL2-Luc-*Id2* vector along with 100 ng of either the CTRL plasmid (described above) or a plasmid encoding FLAG-tagged OVOL2 (pEF-OVOL2-FLAG). After 48 h, cells were lysed using the Bright-Glo Luciferase Assay System (E2610, Promega) and luciferase activity was measured using a luminescence microplate reader.

### 2.12. Statistical Analysis

To quantify placental depth and area, immunohistochemistry was conducted on *Ovol2^+/+^, Ovol2^+/^* and *Ovol2^-/-^* placental sections to detect cytokeratin-positive trophoblasts. Images were imported into ImageJ (Version 1.50i) [23], and analyzed. Placental area was measured by capturing the number of pixels in the image that were positive for cytokeratin. Placental depth was measured by determining the number of pixels in a straight line from the bottom of the placenta (chorionic plate) to the top. Each placenta was measured three independent times, and measurements averaged. Statistical comparisons between two means were tested using Student’s *t*-test and statistical comparisons between three or more means were tested using Analysis of Variance, followed by a Tukey’s post-hoc test. Means were considered statistically different if P was less than 0.05. GraphPad Prism 6.0 was used for all graphing and statistical analysis. All experiments were repeated at least three independent times.

## 3. Results

### 3.1. OVOL2 Is Highly Expressed in Mouse Placenta

To analyze expression of *Ovol2* in the developing mouse conceptus at various times during gestation, RT-PCR was performed on placenta and embryos at E9.5, E12.5, and E15.5. On E18.5, *Ovol2* expression in placenta and various fetal organs (liver, skin, heart, brain, intestine) was evaluated. *Ovol2* was highly expressed in placenta at all time points. In embryos, we were unable to detect expression on E9.5, 12.5, and 15.5 at the PCR conditions used in this study. On E18.5, *Ovol2* was detectable in skin, but was low or undetectable in liver, heart, intestine, and brain (Figure 1A). Similar results were obtained using quantitative RT-PCR, in which the highest expression levels were found in placenta, followed by fetal skin (Appendix A). When assessed at the protein level, OVOL2 was highly expressed in mouse placenta throughout gestation (Figure 1B; note that the lower band likely represents a degradation product, as shown in [24]. Verification that antibody detects OVOL2 is shown in Appendix A). The molecular weight of OVOL2 in mouse placenta was approximately 37 kDa, which is consistent with the OVOL2A isoform that has transcriptional repressor activity [10]. Next, to determine localization of *Ovol2* in mouse placenta, we performed in situ hybridization on serial sections of mouse placenta throughout gestation. *Ovol2* was strongly detected in the chorioallantoic placenta at E9.5, and in the labyrinth zone in E12.5, E15.5, and E18.5 placentas (Figure 1C; data only shown for E9.5 and E12.5. Cytokeratin was used to identify trophoblasts). Expression of *Ovol2* was low or undetectable in the decidua, junctional zone, and chorionic plate at all time points.

### 3.2. Ovol2-Null Placentas Exhibit Developmental Impairments with Evidence of Poor Trophoblast Differentiation

To determine the importance of OVOL2 for placental development, *Ovol2*-heterozygous male and female mice were bred to obtain wild-type, *Ovol2*-heterozygous, and *Ovol2*-null placentas. On E9.5 (prior to embryonic death on or after E10.5), *Ovol2*-null placentas were readily detectable, but were already smaller, paler, and poorly formed when compared to their wild-type siblings (Figure 2A). Using cytokeratin as a marker for all trophoblast lineages, we determined that *Ovol2*-null placentas have significantly reduced depth and area (65% and 63% decreased compared to wild-type placentas, Figure 2B, N = 4, *P* < 0.05). Since OVOL factors control differentiation of various epithelial lineages, we next evaluated expression of various genes associated with trophoblast stem and differentiation states in wild-type, *Ovol2*-heterozygous, and *Ovol2*-null placentas on E9.5. TS cell stem markers *Id2*, *Esrrb*, and *Eomes* were significantly upregulated in *Ovol2-*null placentas by 36%, 63%, and 77%, respectively, when compared to wild-type placentas (Figure 2C, N = 3, *P* < 0.05). There was no change in expression of the TS cell marker *Cdx2* between wild-type and *Ovol2*-null placentas, indicating that only selected stem-related transcripts are up-regulated in placentas lacking OVOL2. *Ovol2-*null placentas also showed reduced expression of labyrinth zone markers (78% and 68% decreased *Synb* and *Gcm1* expression, respectively) and junctional zone-associated transcripts (68% and 82% decreased *Prl3b1* and *Tpbpa* expression, respectively, Figure 2C, all N = 3, *P* < 0.05). Lastly, we performed immunohistochemistry for TPBPA, a protein expressed by a subset of differentiated trophoblasts. In comparison to robust expression of TPBPA evident in wild-type placentas, TPBPA was barely detectable in *Ovol2*-null placentas, suggesting poor trophoblast differentiation in *Ovol2*-null conceptuses (representative images of TPBPA immunohistochemistry conducted on four wild-type and *Ovol2*-null placentas is presented in Figure 2D). Collectively, these results indicate that *Ovol2*-null placentas have impaired placental development with evidence of poor trophoblast differentiation.

### 3.3. Ovol2 Is Induced during Differentiation of Mouse TS Cells

Next, we evaluated expression patterns of OVOL2 during mouse TS cell differentiation. To confirm the efficiency of our TS cell differentiation protocol, we performed quantitative RT-PCR to detect expression of various genes associated with stem and differentiated states in TS cells cultured under stem conditions or following differentiation for up to 14 days. Differentiation of TS cells was associated with a rapid reduction in expression of stem-state associated genes *Cdx2*, *Eomes*, *Esrrb*, and *Id2* (approximately 90-fold reduction in expression of these genes after culturing for seven days in differentiation conditions compared to stem conditions, Figure 3A, N = 6, *P* < 0.05), and increased expression of differentiation markers associated with the labyrinth zone (*Gcm1* and *Synb*; upregulated by 180-fold and 10-fold, respectively; Figure 3B, N = 6, *P* < 0.05) and junctional zone (*Prl3b1* and *Tpbpa*; upregulated by 5824-fold and 4370-fold, respectively; Figure 3C, N = 6, *P* < 0.05). Interestingly, TS cells exposed to differentiation conditions showed a 40-fold upregulation of *Ovol2* transcript when compared to TS cells cultured in stem conditions (Figure 3D, N = 6, *P* < 0.05). OVOL2 protein was detected in both stem and differentiation conditions, but expression was more robust following TS cell differentiation (Figure 3E), which is consistent with the increased expression of *Ovol2* mRNA.

Although mouse TS cells differentiate into all trophoblast lineages, the standard differentiation protocol (removal of FGF4, heparin, activin A, and MEF-conditioned media) preferentially induces trophoblast giant cell formation. Therefore, we used another differentiation protocol, in which TS cells were exposed to differentiation conditions in the presence of the GSK3β inhibitor CHIR99021, which stimulates TS cells to differentiate towards the syncytiotrophoblast II lineage [20]. Following four days of TS cell differentiation in the presence of CHIR99021, stem markers *Cdx2* and *Eomes* were significantly decreased (91% and 92% decrease compared to untreated controls, Appendix A, N = 3, *P* < 0.05). The junctional zone-associated transcript, *Prl3b1*, was significantly upregulated by 93% when compared to the untreated control, which was comparable to the expression of *Prl3b1* following TS cell differentiation for four days without CHIR99021 (Appendix A, N = 3, *P* < 0.05). When syncytiotrophoblast layer II markers were analyzed, expression of *Gcm1* and *Synb* increased in cells cultured in differentiation conditions for four days (5.4-fold and 16.3-fold increased expression compared to TS cells cultured in stem conditions, respectively; Appendix A, N = 3, *P* < 0.05). However, when TS cells were cultured in differentiation conditions in the presence of CHIR99021, expression of *Gcm1* and *Synb* was much more pronounced (225-fold and 2444-fold increased expression compared to TS cells cultured in stem conditions, respectively, Appendix A, N = 3, *P* < 0.05). On the other hand, *Syna*, a marker of syncytiotrophoblast layer I, was upregulated by 82.7-fold in TS cells differentiated for four days under standard differentiation conditions, but was only upregulated by 7.2-fold in differentiated TS cells treated with CHIR99021, when compared to undifferentiated TS cells (Appendix A, N = 3, *P* < 0.05). These results show that CHIR99021 facilitates TS cell differentiation toward the syncytiotrophoblast II lineage. When we analyzed *Ovol2* expression, in comparison to TS cells cultured in stem conditions, both the standard differentiation protocol and addition of CHIR99021 upregulated *Ovol2* expression by approximately 7-fold after four days in comparison to undifferentiated TS cells (Appendix A, N = 3, *P* < 0.05). These results suggest that both TS cell differentiation protocols induce *Ovol2* expression. Therefore, for the duration of this study, only the standard differentiation protocol was used for experiments.

### 3.4. Ectopic Expression of OVOL2 Drives Precocious TS Cell Differentiation

Since *Ovol2* expression increased during TS cell differentiation, we next determined whether increased expression of OVOL2 was sufficient per se to drive differentiation of mouse TS cells. We transfected TS cells with either a CTRL plasmid, or a plasmid encoding OVOL2 with a C-terminal FLAG tag. Mouse TS cells ectopically expressing OVOL2-FLAG showed a 54-fold increase in *Ovol2* expression compared to CTRL cells cultured in stem conditions (Figure 4A, N = 3, *P* < 0.05). Ectopic OVOL2-FLAG expression was also confirmed at the protein level (Figure 4B). Since higher expression of OVOL2 was apparent in TS cells transfected with OVOL2-FLAG, these cells will henceforth be referred to as OVOL2 overexpression (OVOL2 OE) cells. To determine whether increased OVOL2 drives TS cell differentiation, we analyzed expression of genes associated with the trophoblast stem state (*Cdx2, Eomes,* and *Id2*), as well as labyrinth and junctional zone markers (*Gcm1*, *Syna*, *Synb*, *Prl3b1, Tpbpa*) in CTRL and OVOL2 OE cells cultured in the stem state, or following a 1-day exposure to differentiation conditions. OVOL2 OE cells had reduced *Eomes* and *Id2* expression in stem conditions by 53% and 36%, respectively, when compared to CTRL cells (Figure 4C, N = 3, *P* < 0.05). No change was apparent in *Eomes* and *Id2* transcript expression between CTRL and OVOL2 at 1 day of differentiation (Figure 4C, N = 3). *Cdx2* showed no change between CTRL and OVOL2 OE cells at either time point (data not shown). We next assayed expression of genes associated with differentiation of TS cells. Compared to CTRL cells, OVOL2 OE cells had increased *Synb* expression when cultured in the stem state (2.8-fold increase, Figure 4D, N = 3, *P* < 0.05), and increased expression of *Gcm1*, *Synb*, and *Prl3b1* following 1-day culture in differentiation conditions (2.9-fold, 3.6-fold, and 1.8-fold increased expression compared to CTRL cells cultured for 1 day in differentiation conditions, respectively, Figure 4D, N = 3, *P* < 0.05). There was no change in expression of *Syna* (Figure 4D) or *Tpbpa* (not shown) between control and OVOL2 OE cells at either time point. These data suggest that ectopic expression of OVOL2 is sufficient to prime TS cell differentiation, by repressing a subset of stem state genes and inducing expression of a subset of genes associated with differentiation.

### 3.5. Ovol2-Null TS Cells Exhibit Increased Proliferation and Impaired Differentiation Capacity

We next generated *Ovol2*-null TS cells to determine the requirement of OVOL2 for TS cell differentiation. To generate *Ovol2*-null TS cells, blastocysts from male and female *Ovol2-*heterozygous matings were collected, cultured on MEFs, and disaggregated several times to promote formation of TS cells. Wild-type and *Ovol2*-heterozygous TS cells formed tight colonies with morphologies consistent with typical TS cell behavior. *Ovol2*-null TS cells, on the other hand, formed enormous TS cell aggregates that seemed to proliferate approximately twice as fast compared to wild-type or *Ovol2*-heterozygote TS cells (Figure 5, phase-contrast images). To evaluate TS cell proliferative potential, we performed an EdU incorporation assay. In comparison to both wild-type and *Ovol2-*heterozygous TS cells, a greater percentage of *Ovol2*-null TS cells incorporated EdU (approximately 2-fold increased EdU incorporation compared to wild-type and *Ovol2*-heterozygous TS cells, respectively, Figure 5, N = 3, *P* < 0.05), indicating that *Ovol2*-null TS cells proliferated at a faster rate than wild-type or *Ovol2*-heterozygous TS cells. Since wild-type and *Ovol2-*heterozygous TS cells had similar morphologies and levels of proliferation, subsequent experiments were conducted using *Ovol2-*heterozygous and *Ovol2*-null TS cells.

When cultured under stem conditions, we found *Ovol2-*null TS cells expressed higher levels of TS cell stem markers *Id2*, *Esrrb*, and *Eomes* (5.9-fold, 2.9-fold, and 9.9-fold increased expression compared to *Ovol2*-heterozygous TS cells, Figure 6A, N = 3, *P* < 0.05). Culturing *Ovol2*-heterozygous and *Ovol2*-null TS cells in differentiation conditions repressed expression of *Id2*, *Esrrb*, and *Eomes* relative to culturing in their respective stem conditions, but *Ovol2*-null TS cells consistently maintained high levels of these genes. Thus, compared to *Ovol2*-heterozygous TS cells, *Id2*, *Esrrb*, and *Eomes* were upregulated in *Ovol2*-null TS cells at all differentiation time points (Figure 6A, N = 3, *P* < 0.05). In both *Ovol2-*heterozygote and null TS cells, high levels of *Cdx2* were observed in stem conditions, and expression progressively decreased throughout differentiation. During differentiation of *Ovol2*-heterozygous TS cells, there was a robust, time-dependent induction of various differentiation markers including *Gcm1*, *Syna*, *Prl3b1*, and *Ascl2*. Strikingly, *Ovol2*-null TS cells exhibited poor expression of these differentiation-associated genes at all time points. Following culture in differentiation conditions for five days, when *Gcm1* expression peaked in *Ovol2-*heterozygous TS cells, *Ovol2*-null TS cells had a 12-fold decreased expression (Figure 6B, N = 3, all *P* < 0.05). Likewise, following culture in differentiation conditions for 10 days, *Ovol2*-null TS cells exhibited a 61-fold, 9.2-fold, and 11-fold decreased expression of *Syna*, *Prl3b1*, and *Ascl2* compared to *Ovol2*-heterozygous TS cells, respectively (when these genes were most highly expressed in *Ovol2*-heterozygotes, Figure 6B, N = 3, all *P* < 0.05). These results indicate that *Ovol2*-null TS cells are arrested in the stem state and are incapable of differentiating into the various lineages of the mouse placenta.

### 3.6. OVOL2 Binds Upstream of Id2 and Represses Its Activity

We next performed ChIP to evaluate putative transcriptional targets of OVOL2. Since OVOL2 is a transcriptional repressor, we restricted our analysis to regions near target genes exhibiting expression patterns that were consistently upregulated in both *Ovol2*-null placentas and *Ovol2-*null TS cells, which limited our analysis to three putative targets: *Esrrb, Eomes,* and *Id2*. Through in silico analysis of the promoter regions of each gene, we identified putative OVOL2 binding sites based on the presence of known hexameric DNA binding sites (CCGTTA): *Esrrb*: 1477 bp upstream of the transcription start site, *Eomes:* 3193 bp upstream of the transcription start site, and *Id2*: 25 bp downstream of the transcription start site. Using wild-type TS cells, we were unable to detect OVOL2 binding upstream of either *Eomes* or *Esrrb*, but detected robust binding within *Id2* (Figure 7A, N = 3, *P* < 0.05), indicating that the *Id2* gene is a direct transcriptional target of OVOL2. Subsequently, we cloned a sequence of the mouse *Id2* promoter (including the OVOL2 binding region) upstream of firefly luciferase and transfected this vector into HEK-293T cells along with either the CTRL or OVOL2-FLAG expression plasmids. In cells transfected with OVOL2-FLAG, the ability of the *Id2* promoter to drive luciferase activity was reduced by 87% compared to cells transfected with the CTRL vector, strongly suggesting that OVOL2 represses the activity of the *Id2* promoter (Figure 7B, N = 3, *P* < 0.05). Finally, to determine whether ID2 is critical for promoting the stem state in TS cells, we ectopically expressed *Id2* in mouse TS cells. Ectopic expression of *Id2* elicited a 2.7-fold increased expression of *Id2* compared to TS cells transfected with the CTRL plasmid (Figure 7C, N = 3, *P* < 0.05). TS cells expressing increased *Id2* did not exhibit changes in expression of *Cdx2* or various differentiation markers but displayed a 1.5-fold and 1.7-fold increased expression in *Eomes* and *Esrrb*, respectively (Figure 7C, N = 3, *P* < 0.05). Our results suggest that ID2 enhances expression of *Eomes* and *Esrrb*, thereby reinforcing the TS cell stem state.

## 4. Discussion

Differentiation of TS cells into functionally distinct trophoblast lineages is fundamentally important for proper development of the placenta. In this study, we show that OVOL2 is highly expressed in mouse placenta, and is a critical regulator of TS cell differentiation, trophoblast development, and placental morphogenesis. Placentas and TS cells lacking OVOL2 expressed higher levels of trophoblast stem-related genes and showed poor differentiation potential, suggesting that OVOL2 is required to transition TS cells to specialized trophoblast subtypes during mouse placental development. Our results also suggest that OVOL2 represses the TS cell stem state, at least in-part, by controlling expression of the gene encoding ID2. Collectively, our results reveal new insights into the control of TS cell fate during placental development.

In our first series of experiments, we investigated expression and localization patterns of OVOL2 in mouse conceptuses (including decidua, placenta, and embryo) throughout gestation. We detected high expression of *Ovol2* in the labyrinth zone of the placenta throughout gestation, which is consistent with other studies [18,19,25]. In early gestation (E9.5–E15.5) embryos, *Ovol2* expression was low or undetectable at the PCR conditions used in our study, although we cannot conclude that it is not expressed. A previous study showed low expression levels of OVOL2 in embryonic tissue throughout gestation [18]. In E18.5 embryos, expression of *Ovol2* was readily detected in skin but was low in heart, liver, intestine, and brain, which is consistent with other reports describing its restricted expression patterns [10]. The high expression of OVOL2 in placenta throughout gestation suggests that this transcription factor has an important role in placental development and/or function. Indeed, mice lacking OVOL2 exhibit embryonic lethality around E10.5 associated with major defects in blood vessel expansion in the labyrinth zone, placental formation, and heart development [18]. When analyzing conceptuses on E9.5, we observed that *Ovol2*-deficient placentas were noticeably smaller, paler, and exhibited decreased area and depth compared to wild-type placentas. Interestingly, compared to wild-type placentas*, Ovol2-*null placentas exhibited elevated expression of several genes associated with the TS cell stem state, including *Id2*, *Eomes*, *and Esrrb*. Notably, there was no difference in *Cdx2* expression, indicating that *Ovol2* deficiency likely only impacts expression of a subset of trophoblast stem-related genes. We also found decreased expression of various genes associated with differentiated trophoblasts (*Gcm1*, *Prl3b1*, *Synb*, and *Tpbpa*) in *Ovol2*-null placentas compared to wild-type placentas. It should be noted that previous studies detected *Tpbpa* and *Gcm1* in *Ovol2*-null placentas [18] and concluded that expression was similar to *Ovol2-*heterozygous placentas, although these experiments were conducted using in situ hybridization so expression levels of these genes were not quantified. Our immunohistochemical analysis of TPBPA, a marker of differentiated trophoblasts populating the junctional zone [9], shows a strong reduction in protein expression in *Ovol2*-null placentas, suggesting that these placentas may lack developmental cues required for trophoblast differentiation. Since *Ovol2-*null placentas develop poorly and exhibit gene expression profiles consistent with poor trophoblast development, we next investigated the role of OVOL2 in the control of TS cell function.

Mouse TS cells can be derived from preimplantation blastocysts or extraembryonic ectoderm and propagated in vitro; they represent the precursor cells of all specialized trophoblast lineages. Since culture conditions that favor mouse TS cell self-renewal or differentiation have been known for decades, these cells have provided the best studied in vitro models to investigate development of trophoblast lineages. Using two separate differentiation protocols, we induced mouse TS cells to either undergo conventional differentiation [4], or differentiate to preferentially form cells representative of the syncytiotrophoblast layer II lineage [21]. We found that *Ovol2* mRNA levels robustly increased during both TS cell differentiation protocols. OVOL2 protein was detectable in TS cells cultured in the stem state, and expression increased following induction of TS cell differentiation. Moreover, we found that ectopic expression of OVOL2 was sufficient to prime TS cells to undergo precocious differentiation, including decreased levels of stem-related genes *Id2* and *Eomes*, and increased expression of *Gcm1*, *Synb*, and *Prl3b1.* However, not all differentiation-associated genes (e.g., *Syna*) were induced in OVOL2 OE cells, indicating that OVOL2 may not be sufficient by itself to stimulate full-fledged TS cell differentiation. This trend was observed at early stages of differentiation but was imperceptible by three and five days of differentiation (not shown), likely relating to high levels of endogenous OVOL2 in differentiating trophoblasts by these time points. Interestingly, TS cells lacking the transcriptional regulator lysine-specific demethylase 1 exhibit precocious trophoblast differentiation and migration attributed to increased expression of *Ovol2*, further suggesting that increased OVOL2 is sufficient to drive TS cell differentiation [26].

To determine whether OVOL2 is required for TS cell differentiation, we generated TS cells from *Ovol2-*heterozygous and *Ovol2*-null blastocysts. *Ovol2-*heterozygous TS cells formed classic TS cell colonies that were morphologically indistinguishable from wild-type TS cells. On the other hand, *Ovol2*-null TS cells seemed to proliferate vigorously, and required passaging at a rate two-to-three times more frequently than wild-type or heterozygous TS cells. When we assessed levels of trophoblast stem-related genes, *Ovol2*-null TS cells exhibited much higher levels of *Eomes*, *Esrrb*, and *Id2* than *Ovol2-*heterozygous TS cells. These results suggest that in the absence of OVOL2, TS cells acquire more proficient stem-like properties, which may explain the higher proliferative index in these cells. Our findings that OVOL2 suppresses TS cell proliferation are consistent with the known tumor suppressor functions of OVOL2 in other cell types [27]. OVOL2 inhibits cell proliferation and protects epithelial identity by regulating Wnt signaling [14,28,29,30], repressing oncogenes (e.g., *c-Myc*) [24], and reducing expression of genes associated with epithelial-to-mesenchymal transition [16,27,31,32,33,34]. Interestingly, in our study, expression of *Eomes*, *Esrrb*, and *Id2* remained higher in *Ovol2-*null TS cells even when cultured in differentiation conditions, resulting in diminished capacity to upregulate differentiation markers including *Gcm1*, *Syna*, *Prl3b1*, and *Ascl2*. The poor differentiation potential of TS cells lacking OVOL2 is likely responsible for failure of proper morphogenesis in *Ovol2*-deficient placentas, and may also contribute to the poor vascularization in the labyrinth zone that was described previously [18].

OVOL2A contains an N-terminal SNAG repressor motif and four C-terminal C2H2 zinc finger domains that confer DNA binding activity [10,12], and acts as a transcriptional repressor [16]. Therefore, to identify putative transcriptional targets of OVOL2 binding, we performed in silico analysis of OVOL2 consensus sequences in the proximal promoter regions of three genes that consistently showed increased expression in *Ovol2-*deficient placentas and TS cells: *Eomes, Esrrb,* and *Id2.* While putative ovo binding motifs were detected near all three genes, we only detected OVOL2 binding within *Id2,* resulting in repression of *Id2* promoter activity. *Id2* encodes a dominant-negative helix-loop-helix protein that prevents cell differentiation in numerous developmental pathways by preventing DNA binding of other basic helix-loop-helix transcription factors [35,36]. *Id2* has previously been identified as a target of OVOL1-mediated repression during progression through the pachytene stage of spermatogenesis. Similar to our findings in *Ovol2-*deficient trophoblasts, *Ovol1*-deficient spermatogonia express high levels of *Id2*, and exhibit a reduced capacity to exit the cell cycle and commence differentiation [37]. Using a published ChIP-Sequencing dataset [16], we also detected OVOL2 enrichment near the *Id2* gene in mouse mammary epithelial cells, suggesting that OVOL factors may regulate *Id2* transcription in a variety of tissues. Although mice lacking ID2 are viable and fertile, ID1 and ID3 are also expressed in mouse placenta. Mice lacking multiple ID proteins are embryonic lethal, suggesting compensatory actions of other ID family members (reviewed in [38]). Studies have not yet investigated whether placental defects are evident in mice lacking multiple ID isoforms. However, ID2 is the earliest transcriptional regulator specifically expressed in the nascent trophectoderm, suggesting a critical role for ID2 in maintenance of the TS cell state ([39]). The importance of ID2 repression for trophoblast differentiation has been previously demonstrated in the mouse SM10 labyrinth trophoblast progenitor cell line, in which overexpression of ID2 prevents differentiation [40]. Expression of ID2 is also highly expressed in human cytotrophoblasts, and ID2 expression decreases during differentiation in vitro and in situ. Overexpression of ID2 is sufficient to prevent differentiation of human cytotrophoblasts and cell-lines [41,42,43]. Consistent with these studies, we found that ectopic expression of ID2 in TS cells was sufficient to drive increased levels of stem state-associated transcripts. Although it is likely that *Id2* is not the only transcriptional target of OVOL2-mediated repression in TS cells, our results strongly implicate repression of *Id2* as a mechanism through which OVOL2 transitions these cells towards a differentiated phenotype.

In conclusion, we identified a critical function of OVOL2 for the progression of TS cell differentiation during placental morphogenesis. OVOL2 represses *Id2* and appears to serve as a gatekeeper that enables TS cells to switch from a stem state to a differentiated phenotype. In essence, OVOL2 serves a role in mouse trophoblast development that parallels the function of OVOL1 in the promotion of human cytotrophoblast differentiation [19]. In future studies, it would be interesting to determine whether other genes besides *Id2* are targeted for repression by OVOL2 during TS cell differentiation, and also identify how ID2 functions to reinforce the TS cell stem state. Given the high expression of OVOL2 in mouse placenta and compelling effects on TS cell behavior, it is tempting to speculate that failure of TS cell differentiation during placental morphogenesis could contribute, at least in part, to the vascular deficiencies and embryonic lethality observed in *Ovol2*-deficient embryos.

## Figures and Tables

**Figure 1 cells-09-00840-f001:**
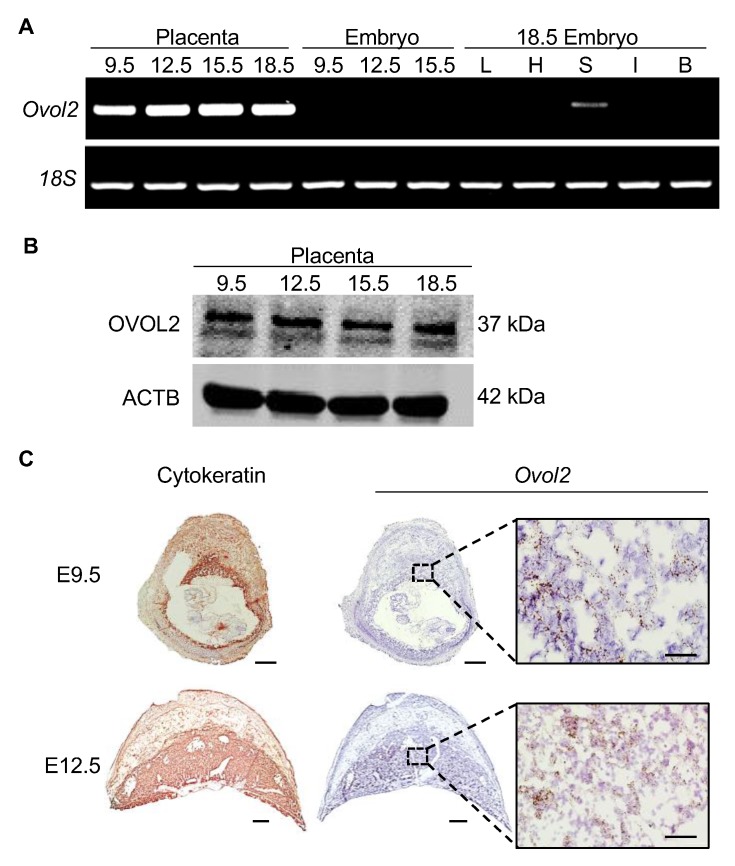
OVOL2 expression in mouse tissue throughout gestation. (**A**) RT-PCR analysis showing expression of *Ovol2* and *Rn18s* (*18S*) in mouse placenta and embryo on E9.5, 12.5, 15.5, and 18.5. *Ovol2* expression in embryonic liver (L), heart (H), skin (S), intestine (I), and brain (B) were also assessed on E18.5. (**B**) Western blot analysis showing protein expression of OVOL2 in mouse placenta at E9.5, 12.5, 15.5, and 18.5. ACTB was used as a loading control. (**C**) In situ hybridization showing *Ovol2* transcript localization in mouse placenta at E9.5 and 12.5. Cytokeratin immunohistochemistry was used to delineate the placenta. The dashed boxes in the low magnification images denote the location of the high magnification images. Scale bar = 500 µm for all images except for the high magnification *Ovol2* panels, which represent 50 µm.

**Figure 2 cells-09-00840-f002:**
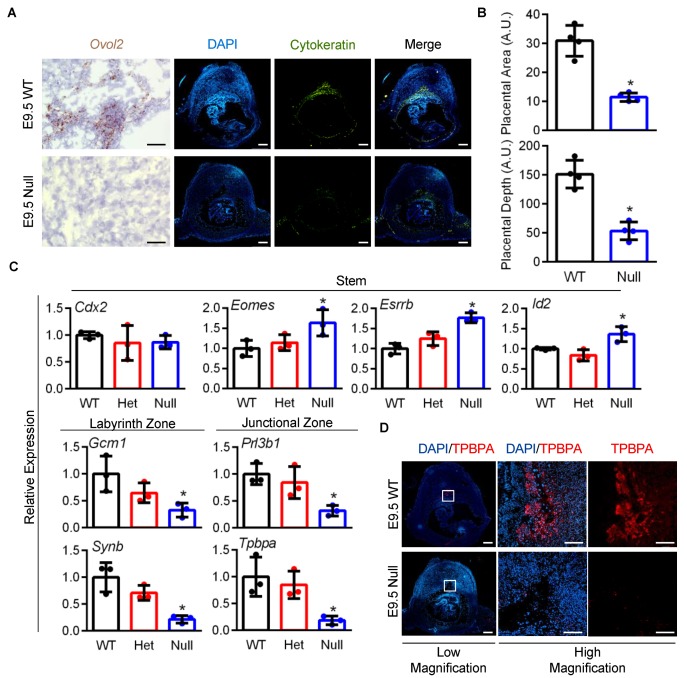
Placental development is impaired in *Ovol2*-deficient embryos. (**A**) Evaluation of *Ovol2* expression in wild-type (WT) and *Ovol2-*null placentas on E9.5. Immunohistochemical staining of cytokeratin was used to denote the location of placentas. Note poor placental development in *Ovol2-*null placentas. (**B**) Quantitative analysis of placental area and depth comparing WT and *Ovol2-*null placentas. (**C**) Transcript analysis of stem (*Cdx2, Eomes, Esrrb*, and *Id2*) and differentiation markers (*Gcm1*, *Synb*, *Prl3b1*, and *Tpbpa*) comparing WT, heterozygote (Het), and *Ovol2*-null placentas at E9.5. (**D**) Immunohistochemical staining of differentiation marker TPBPA in WT and *Ovol2*-null placentas. Values significantly different from WT (N = 3–4 from different dams, *P* < 0.05) are denoted with an asterisk (*). Scale bars = 500 µm (low magnification) and 100 µm (high magnification). Graphs represent means (SD).

**Figure 3 cells-09-00840-f003:**
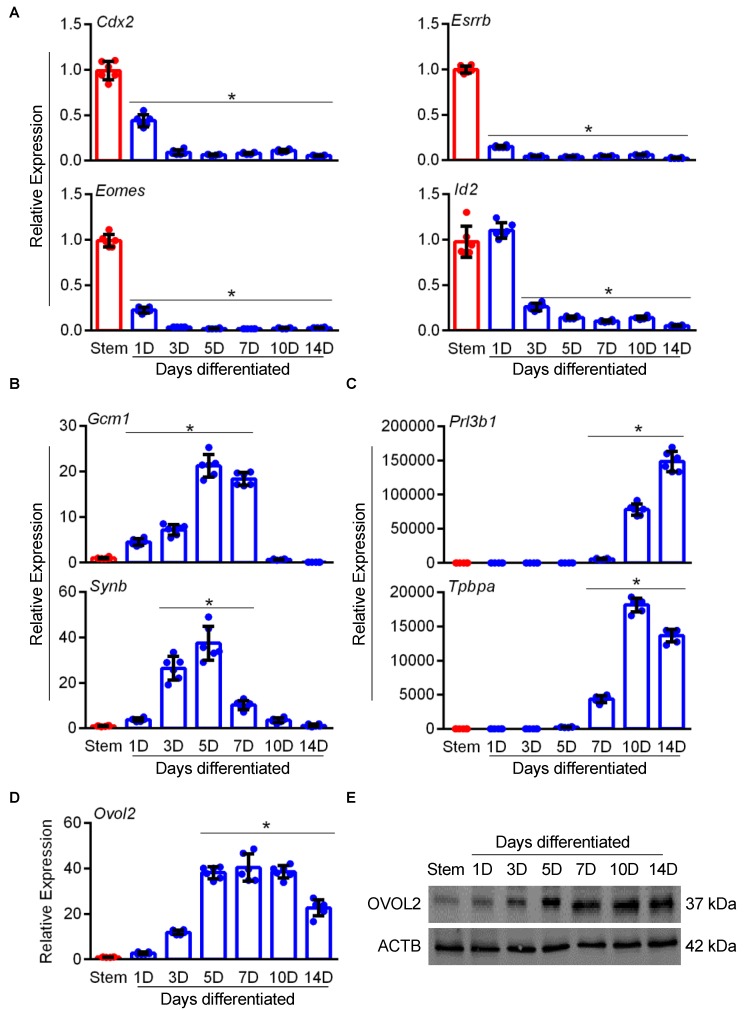
Increased expression of OVOL2 during mouse TS cell differentiation. Quantitative RT-PCR analysis of (**A**) stem (*Cdx2, Eomes*, *Esrrb*, and *Id2*) and differentiation markers for the (**B**) labyrinth zone (*Gcm1*, *Synb*) and (**C**) junctional zone (*Prl3b1* and *Tpbpa*) in mouse TS cells cultured under stem or differentiation conditions for up to 14 days. (**D**) Quantitative RT-PCR and (**E**) and Western blot analysis of OVOL2 expression in mouse TS cells cultured under stem or differentiation conditions for up to 14 days. ACTB was used as a loading control for Western blots. Values significantly different from stem conditions (N = 6, *P* < 0.05) are indicated with an asterisk (*). Graphs represent means (SD).

**Figure 4 cells-09-00840-f004:**
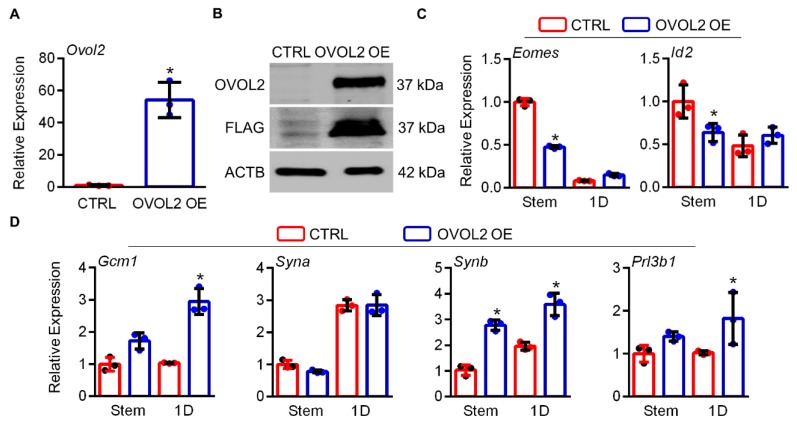
Ectopic expression of OVOL2 stimulates precocious differentiation of mouse TS cells. (**A**) Quantitative RT-PCR to detect *Ovol2* expression in mouse TS cells transfected with either an empty vector (CTRL) or a plasmid encoding OVOL2 fused to a C-terminal FLAG tag (OVOL2 OE). (**B**) Western blot analysis to detect OVOL2 and FLAG in CTRL and OVOL2 OE mouse TS cells. ACTB was used as a loading control. Quantitative RT-PCR for (**C**) stem (*Eomes* and *Id2*) and (**D**) differentiation markers (*Gcm1*, *Syna*, *Synb*, and *Prl3b1*) in CTRL or OVOL2 OE mouse TS cells cultured in stem conditions or following a 1-day (1D) exposure to differentiation conditions. Values significantly different (N = 3, *P* < 0.05) are indicated with an asterisk (*). Graphs represent means (SD).

**Figure 5 cells-09-00840-f005:**
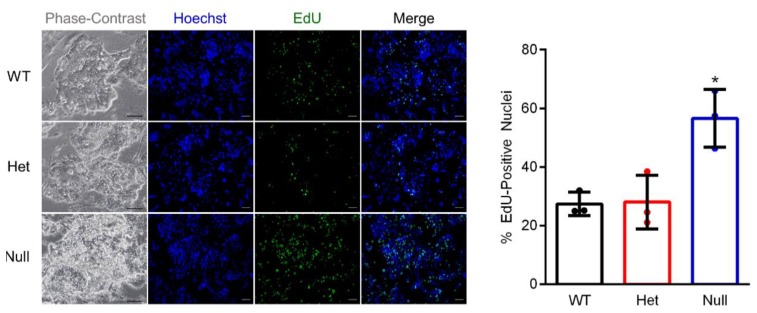
*Ovol2*-null TS cells exhibit increased proliferation. Representative phase-contrast images (left panels) of wild-type (WT), *Ovol2*-heterozygote (Het), and *Ovol2*-null mouse TS cells. Detection of incorporated EdU (green) was used to assess proliferation of cells. Hoechst was used to detect nuclei (blue). The percent EdU-positive nuclei is shown in the graph on the right. Values significantly different from WT (N = 3, *P* < 0.05) are indicated with an asterisk (*). Scale bars = 100 µm. Graph represents means (SD).

**Figure 6 cells-09-00840-f006:**
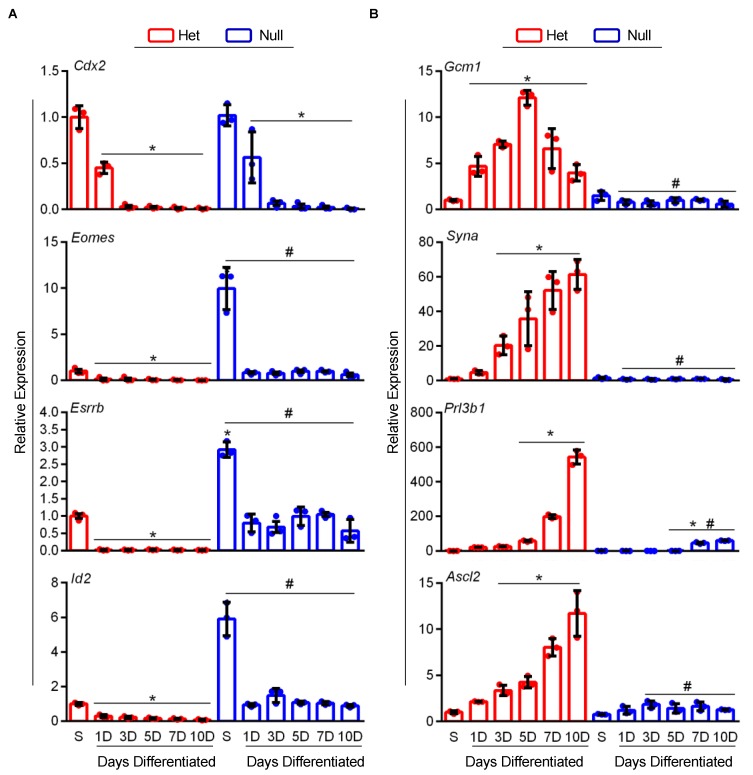
*Ovol2*-null TS cells express high levels of stem-related genes and exhibit impaired differentiation potential. Quantitative RT-PCR was used to analyze expression of (**A**) stem (*Cdx2*, *Esrrb*, *Eomes*, and *Id2*) and (**B**) differentiation markers (*Gcm1*, *Syna*, *Prl3b1*, and *Ascl2*) in *Ovol*2-heterozygote (Het) and null TS cells cultured in stem conditions (S) or up to 10 days in differentiation conditions. Values significantly different from *Ovol2*-Het TS cells cultured in stem conditions are represented with an asterisk (*, N = 3, *P* < 0.05); values significantly different between *Ovol2*-Het and null TS cells at each respective time point are indicated with a number sign (#, *P* < 0.05). Graphs represent means (SD).

**Figure 7 cells-09-00840-f007:**
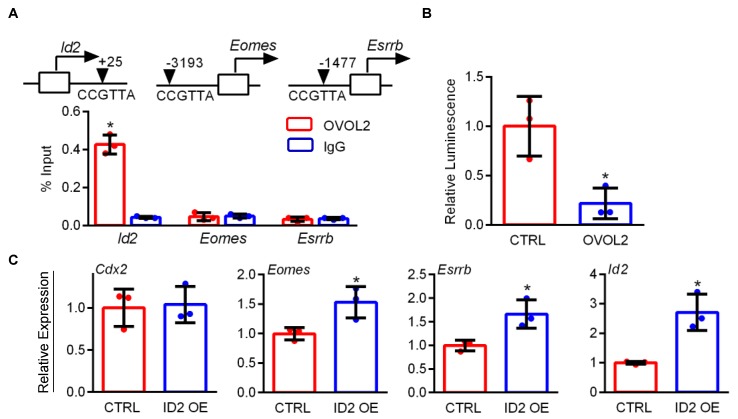
OVOL2 interacts with the *Id2* gene. (**A**) ChIP was performed using OVOL2 or IgG (negative control) antibodies in lysates of mouse TS cells differentiated for 3 days. Quantitative PCR was used to detect enrichment, using primers spanning CCGTTA sequences near transcription start sites of *Id2*, *Eomes*, and *Esrrb*. (**B**) Relative luciferase activity measured in HEK-293T cells transfected with *Id2-Luc* and either the CTRL plasmid or pEF-OVOL2-FLAG. (**C**) Quantitative RT-PCR showing relative expression of *Cdx2*, *Eomes*, *Esrrb*, and *Id2* in mouse TS cells transfected with CTRL or pEF-ID2-FLAG (ID2 OE). Values significantly different (N = 3, *P* < 0.05) from controls are indicated with an asterisk (*). Graphs represent means (SD).

**Table 1 cells-09-00840-t001:** Forward and reverse primers used for RT-PCR and quantitative RT-PCR amplification.

Gene	Accession No.	Primers	Product Size
*Ascl2*	NM_008554.3	FWD: 5′-TGGAAGCACACCTTGACTGG-3′ REV: 5′-TGGACGTTTGCACCTTCAC-3′	114
*Cdx2*	NM_007673.3	FWD: 5′-CCAGCTCTTTGCCTCTCTGT-3′ REV: 5′-TGCCTCTGGCTCCTGTAGTT-3′	175
*Eomes*	NM_001164789.1	FWD: 5′-AACATGCAGGGCAATAAGATG-3′ REV: 5′-AGCCTCGGTTGGTATTTGTG-3′	172
*Esrrb*	NM_001159500.1	FWD: 5′-GGGAGCTTGTGTTCCTCATC-3′ REV: 5′-CTACCAGGCGAGAGTGTTCC-3′	200
*Gcm1*	NM_008103.3	FWD: 5′-AACACCAACAACCACAACTCC-3′ REV: 5′-CAGCTTTTCCTCTGCTGCTT-3′	151
*Id2*	NM_010496.3	FWD: 5′-CAGCATCCCCCAGAACAA-3′ REV: 5′-TCTGGTGATGCAGGCTGA-3′	121
*Ovol2* (RT-PCR)	NM_026924.3	FWD: 5′-TGTTCCTTGGAGTCCCACCT-3′ REV: 5′-ACTGACAACTTTGGGGTGCTT-3′	426
*Ovol2* (quantitative RT-PCR)	NM_026924.3	FWD: 5′-TTCACCCAGCGGTGTTCCTT-3′ REV: 5′-TGTAGCCGCAATCCTCACAC-3′	114
*Ovol2* (Genotyping)	NM_026924.3	FWD: 5′-CATAGCCCATGTGTGGCTGCTG-3′ REV: 5′-GCCGGCCTTAAAACATCCCAC-3′	847
*Prl3b1*	NM_008865.3	FWD: 5′-CCAACGTGTGATTGTGGTGT-3′ REV: 5′-TCTTCCGATGTTGTCTGGTG-3′	175
*Syna*	NM_001013751.2	FWD: 5′-CCCTTGTTCCTCTGCCTACTC-3′ REV: 5′-TCATGGGTGTCTCTGTCCAA-3′	175
*Synb*	NM_173420.3	FWD: 5′-TGAACACCCCAACTGAGCAA-3′ REV: 5′-GGACGAAACAGAGGACCCAA-3′	123
*Rn18s*	NR_003278.3	FWD: 5′-GCAATTATTCCCCATGAACG-3′ REV: 5′-GGCCTCACTAAACCATCCAA-3′	122
*Tpbpa*	NM_009411.4	FWD: 5′-CGGAAGGCTCCAACATAGAA-3′ REV: 5′-TCAAATTCAGGGTCATCAACAA-3′	141
*Ywhaz*	NM_001253805.1	FWD: 5′-CCCTCTTGGCAGCTAATGG-3′ REV: 5′-GGAGTAAGGGGAAAATGTGGG-3′	116

**Table 2 cells-09-00840-t002:** Forward and reverse primers used for ChIP.

Gene	CCGTTA	Primers
*Id2*	+25	FWD: 5′-CTTCCTCCTACGAGCAGCAT-3′ REV: 5′-GGTCCGACAGGCTGTTTTT-3′
*Eomes*	−3193	FWD: 5′-TTCCAGAGTCCAGGCAAAGT-3′ REV: 5′-TGCTGCATGCTCTCCTTTTA-3′
*Esrrb*	−1477	FWD: 5′-TTGAGCACCCATTCAAACAC-3′ REV: 5′-ATGCAGGTAAACGCCATGTT-3′

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
