# Peer review of "The Transcription Factor OVOL2 Represses ID2 and Drives Differentiation of Trophoblast Stem Cells and Placental Development in Mice"

_cells, 2020, doi:10.3390/cells9040840_

Round 1

Reviewer 1 Report

Jeyarajah and colleague investigated the functions of Ovol2 in murine placental development. Placentas and trophoblast stem cells lacking Ovol2 had a poor poteitial to differentiate into trophoblast, resulting in increased expression of stem cell-related genes, such as Id2, Essrb and Eomes. Ovol2 repress the expression of Id2 by binding to its promoter regions. Overexpresion of Id2 repressed the differentiation of TS cells. Therefore, they concluded that Ovol2 regulates the differentiation of TS cells by repressing Id2 expression. This study is very interesting. However, some revision is necessary.

  1. Fig.1B; Why two bands were detectable in Ovol2 proteins?
  2. P7-L282; Stem cell-related transcripts were up-regulated?
  3. Fig3E; Expression of Ovol2 proteins was almost constant, even though mRNA expression was markedly increased
  4. Fig5B; Why Ovol2 protein was hardly increased  in OVOL2 OE?
  5. The authors must describe and discuss about trophoblast development in Id2-null mice.
  6. The authors must refer to previous  studies that showed the importance of Id2 gene in  human placenta development.

Author Response

We thank the reviewer for the insightful comments. We have addressed all comments, as follows:

Point 1: Fig.1B; Why two bands were detectable in Ovol2 proteins?

Response 1: The reason why two bands are detected in western blots is an interesting question. In this new version of the manuscript, we include molecular weight indicators for all western blots, and have also included a new supplemental figure (Figure S2) in which we validate the specificity of the antibody used for western blotting. In Figure S2, we show that overexpression of mouse OVOL2 in human embryonic kidney 293T cells produces a strong band at 37 kDa; the same molecular weight as the upper band in Figure 1B. A control vector does not produce any band. Additionally, we used placental lysates from Ovol2 wild-type, heterozygote, and null mice, and show that the antibody produces two bands in wild-type and heterozygote placentas, but no band in null placentas. Since both bands are not present in Ovol2-null placentas, it is not likely that the lower band is a non-specific product. It is possible that two isoforms of the protein are evident in mouse placentas: OVOL2A (~37 kDa), and OVOL2B (~30 kDa), although the lower band is slightly larger than 30 kDa and is not consistently detected. Another likely explanation is that the lower band is a degradation product. This possibility is consistent with Wells et al J Biol Chem 2009, where two bands appear in UG1 cell lysates during western blotting for OVOL2: one at 37 kDa and the other slightly smaller. Wells et al attributed the lower band to a degradation product because its presence varied from experiment to experiment. Since this is similar to what we observed in our experiments, we included this potential explanation in Results lines 269-272.

Point 2: P7-L282; Stem cell-related transcripts were up-regulated?

Response 2: Thank you for the correction. We have revised the sentence to “...indicating that only selected stem-related transcripts are up-regulated in placentas lacking OVOL2.” (Results line 291)

Point 3: Fig3E; Expression of Ovol2 proteins was almost constant, even though mRNA expression was markedly increased

Response 3: We have included a new western blot image in Figure 3E that better parallels the induction of Ovol2 mRNA.

Point 4: Fig5B; Why Ovol2 protein was hardly increased in OVOL2 OE?

Response 4: We agree with the reviewer’s concern. In Figure 5B (now Figure 4B in revised manuscript), we present a new western blot image showing robust OVOL2 and FLAG expression in OVOL2 OE cells, which parallels the high levels of Ovol2 mRNA in these cells.

Point 5: The authors must describe and discuss about trophoblast development in Id2-null mice.

Response 5: We have included a discussion about trophoblast development in Id2 null mice, as follows: “Although mice lacking ID2 are viable and fertile, ID1 and ID3 are also expressed in mouse placenta. Mice lacking multiple ID proteins are embryonic lethal, suggesting compensatory actions of other ID family members (reviewed in [38]). Studies have not yet investigated whether placental defects are evident in mice lacking multiple ID isoforms. However, ID2 is the earliest transcriptional regulator specifically expressed in the nascent trophectoderm, suggesting a critical role for ID2 in maintenance of the TS cell state ([39]). The importance of ID2 repression for trophoblast differentiation has been previously demonstrated in the mouse SM10 labyrinth trophoblast progenitor cell line, in which overexpression of ID2 prevents differentiation [40].”  (Discussion lines 516-522)

Point 6: The authors must refer to previous studies that showed the importance of Id2 gene in  human placenta development.

Response 6: We have included a statement about the importance of ID2 for human trophoblast development, as follows: “Expression of ID2 is also highly expressed in human cytotrophoblasts, and ID2 expression decreases during differentiation in vitro and in situ. Overexpression of ID2 is sufficient to prevent differentiation of human cytotrophoblasts and cell-lines [41-43].” (Discussion lines 523-527)

Reviewer 2 Report

In this study, Jeyarajah et al investigated the molecular mechanism on trophoblast stem (TS) cells differentiation focusing on the role of Ovol2 transcription factor. Authors demonstrated the requirement of Ovol2 in inducing differentiation of TS cells into multiple lineages of placental cells and identified Id2 as a direct downstream mediator of Ovol2-induced TS cell differentiation.

The manuscript is well written and most of conclusions are nicely supported by the data. Although involvement of Ovol2 in TS cell differentiation has been reported, the molecular mechanism described in this study is novel. I recommend this paper to be published after resolving some minor issues as follows.

  1. In Result 3.1 and Figure 3A, authors claimed that Ovol2 was not detected in the embryonic tissue from 9.5 to 15.5. However, an earlier study detected the expression of Ovol2 in embryonic tissues as well as placenta (PMID17573777). Where does this discrepancy come from? I wonder if this is due to differential expression of splicing variants among tissues (PMID). Can the primer sets used in this study detect three major isoforms (A, B, and C) of Ovol2?
  2. In figure 1B, what are the two bands shown in the blot? What are the molecular sizes? It would also be appropriate to show negative and positive controls such as E9.5-15.5 embryos and 18.5 skin, respectively.
  3. In figure 1C, the ISH signal for Ovol2 is not very convincing. Please improve the images. I highly recommend to perform immunohistological analysis for mouse Ovol2 as in previous studies (PMID 24735879/24735878 ).
  4. Figure 4 may be more suitable in supplemental information as this data is not essential for the main story.
  5. In Figure 8B, a luciferase construct for a mutant Ovol2-binding site would be a good control.
  6. Ovol2 overexpression induced trophoblast differentiation markers Gcm1, Synb, and Prl3b1 but not Syna while Ovol2 deletion was required for the induction of all differentiation markers including Syna. How do the authors explain this discrepancy?
  7. Is Id2 a general downstream target of Ovol2 in other epithelial tissues such as skin or mammary gland? A few ChIP-seq data are available for the authors to investigate such as PMID24735879.

Reviewer 3 Report

This is a good manuscript describing an interesting analysis of OVUL2 in placental development. The study reports interesting well-supported findings on a proposed role for OVOL2 in stem cell differentiation to trophoblast lineages. While some aspects of the analysis could be deeper (e.g., target genes of OVOL2 repression), this is a good focused study. I have the following specific comments.

Results

Figures: I think the distribution of data points should be shown in the bar graphs, and I think SD rather than SEM would be preferable.

3.1 and Fig. 1A. Is this qRT-PCR? If not, why not?

Fig.1C OVOL2 staining difficult to see on versions I downloaded.

Fig. 2 How was transcript and protein quantification achieved on sections. I can’t see any detailed description in Methods or Results.

Page 8, line 294. Specify strain /name of TS cells used in Methods or Results section.

Fig. 5 Why was OVOL2 over-expression only analysed at one time point (1 day), and why so soon after transfection?

Page 12, line 366. Do you mean that OVOL2 null TS cells have half (rather than twice) the doubling time of wildtype and heterozygotes? I’m confused; perhaps say that they proliferate twice as fast?

3.6. Why/how were three targets of OVOL2 selected? Are OVOL2 binding sites conserved in human orthologues?

Fig. 8 and ChIP experiments. More clarity about which/how many TS cell lines were used.

Discussion

Perhaps allude to any data on OVOL2 and related pathways and their involvement in human placemta.

Author Response

We thank the reviewer for the insightful comments. We have addressed all comments as follows:

Point 1: Figures: I think the distribution of data points should be shown in the bar graphs, and I think SD rather than SEM would be preferable.

Response 1: As requested, we have changed all bar graphs to dot plots in order to better show the distribution of data points, and have also changed the error bars to represent SD rather than SEM. We are pleased with the appearance of the new graphs and are thankful for this suggestion. 

Point 2: 3.1 and Fig. 1A. Is this qRT-PCR? If not, why not?

Response 2: Figure 1A: This figure represents an image obtained through agarose gel electrophoresis following RT-PCR, not qRT-PCR. We presented data this way to show which tissues OVOL2 was detectable, rather than quantifying relative expression levels between tissues. However, we understand why it may also be valuable to measure expression levels between tissues, so in new Figure S1, we present results of Ovol2 expression in various tissues using qRT-PCR. Consistent with results from RT-PCR, the highest expression levels are detected in placenta, followed by skin. Low expression levels are detectable in other embryonic tissues. The Results section (lines 266-268) has been updated accordingly. 

Point 3: Fig.1C OVOL2 staining difficult to see on versions I downloaded.

Response 3: We apologize that it was difficult to see staining in Fig 1C. We have changed this figure accordingly to make the staining clearer. 

Point 4: Fig. 2 How was transcript and protein quantification achieved on sections. I can’t see any detailed description in Methods or Results.

Response 4: We used cytokeratin immunohistochemistry in order to delineate the placenta on embryonic day 9.5, and then analyzed placental area and thickness. We have included a statement in Methods: statistical analysis referring to quantification of placental area and thickness (Methods lines 248-253).  

Point 5: Page 8, line 294. Specify strain /name of TS cells used in Methods or Results section.

Response 5: We used the F4 mouse TS cell line in our experiments, and have included this detail in Methods (line 132). 

Point 6: Fig. 5 Why was OVOL2 over-expression only analysed at one time point (1 day), and why so soon after transfection?

Response 6: We transfected OVOL2 into mouse TS cells, and analyzed OVOL2 expression in stem conditions and at day 1, 3, and 5. We were interested in determining whether OVOL2 was sufficient to drive TS cell differentiation precociously, which is why we focused on stem and day 1 of differentiation (time points at which differentiation-associated factors are expressed at low or undetectable levels). When we analyzed OVOL2 expression in control and OVOL2 OE cells on day 3 and day 5 of differentiation, differentiation-associated factors were highly upregulated in both control and OVOL2 OE cells, and stem-related factors were decreased. There was also no longer evidence of OVOL2 overexpression in OVOL2 OE cells, likely because of the length of time following transfection, as well as high endogenous Ovol2 expression during TS cell differentiation. We included this detail in Discussion (lines 478-481).

Point 7: Page 12, line 366. Do you mean that OVOL2 null TS cells have half (rather than twice) the doubling time of wildtype and heterozygotes? I’m confused; perhaps say that they proliferate twice as fast?

Response 7: Thank you for the correction. We have changed the sentence to “...that seemed to proliferate approximately twice as fast compared to wild-type or Ovol2-heterozygote TS cells”, as recommended. (Results line 374)

Point 8: 3.6. Why/how were three targets of OVOL2 selected? Are OVOL2 binding sites conserved in human orthologues?

Response 8: Throughout the manuscript, we analyzed four genes associated with the TS cell stem state: Cdx2, Eomes, Esrrb, and Id2. Cdx2 expression was consistently unaffected in placentas and TS cells lacking OVOL2, so we reasoned that Cdx2 is unlikely a transcriptional target of OVOL2. In contrast, Eomes, Esrrb, and Id2 were all consistently up-regulated in placentas and TS cells lacking OVOL2. We also identified OVOL2’s hexameric binding sequence upstream of each gene. This binding site is present in human orthologues of ID2 and EOMES. We further analyzed all three sites to determine whether OVOL2 is enriched at these sites in mouse TS cells cultured in differentiation conditions. However, we only identified OVOL2 binding near Id2. Since ID2 is among the first transcription factors specifically expressed in mouse trophectoderm and inhibits differentiation of both mouse TS cells and human cytotrophoblasts, our data suggest that targeted repression of ID2 by OVOL2 is a critical step for progression of mouse TS cell differentiation. We realize that there may be additional targets of OVOL2 besides Id2 that were not assessed in our study. We have included two sentences in the Discussion to acknowledge the possibility of other factors targeted by OVOL2, as follows: “Although it is likely that Id2 is not the only transcriptional target of OVOL2-mediated repression in TS cells, our results strongly implicate repression of Id2 as a mechanism through which OVOL2 transitions these cells towards a differentiated phenotype.” (Discussion line 528-530); and  “In future studies, it would be interesting to determine whether expression of other genes besides Id2 are directly targeted by OVOL2 during TS cell differentiation.” (Discussion lines 538-540)

Point 9: Fig. 8 and ChIP experiments. More clarity about which/how many TS cell lines were used.

Response 9: We used F4 TS cells to perform ChIP experiments. This information has been added to the Methods section (line 230) and Results (line 412).

Point 10: Discussion: Perhaps allude to any data on OVOL2 and related pathways and their involvement in human placenta.

Response 10: As far as we are aware, there is no information reported about the function of OVOL2 in the human placenta. However, OVOL1 is expressed in human placenta and appears to have a similar role in promoting human trophoblast differentiation as our data show for OVOL2 in the mouse placenta. There is also information available about the role of ID2 in human cytotrophoblast differentiation. We have included several sentences to the Discussion regarding the role of OVOL1 and ID2 in human trophoblast differentiation (Discussion lines 534-536: “In essence, OVOL2 serves a role in mouse trophoblast development that parallels the function of OVOL1 in the promotion of human cytotrophoblast differentiation [19].” and Discussion lines 524-526: “Expression of ID2 is also highly expressed in human cytotrophoblasts, and ID2 expression decreases during differentiation in vitro and in situ. Overexpression of ID2 is sufficient to prevent differentiation of human cytotrophoblasts and cell-lines [41-43].

Round 2

Reviewer 1 Report

The authors succeded to revise the manuscript.